# Effect of Storage Technologies on Postharvest Insect Pest Control and Seed Germination in Mexican Maize Landraces

**DOI:** 10.3390/insects13100878

**Published:** 2022-09-27

**Authors:** Sylvanus Odjo, Nicolas Bongianino, Jessica González Regalado, María Luisa Cabrera Soto, Natalia Palacios-Rojas, Juan Burgueño, Nele Verhulst

**Affiliations:** 1International Maize and Wheat Improvement Center (CIMMYT), Carretera Mexico-Veracruz km 45, El Batán, Texcoco C.P. 56237, Mexico; 2Instituto de Ciencia y Tecnología de Alimentos Córdoba (ICYTAC), CONICET-UNC, Córdoba 5000, Argentina

**Keywords:** hermetic technologies, insects, maize landraces, seed germination, storage losses

## Abstract

**Simple Summary:**

In Mexico, smallholder farmers produce locally adapted maize varieties, generally called landraces, for food. They participate in the conservation of landraces by selecting and storing their seeds from one production cycle to another, but are facing challenges such as storage losses caused by insects (mainly *Sitophilus zeamais* and *Prostephanus truncatus*) and climate change. In the current study, the effectiveness of the conventional storage practices of farmers in the highlands of Mexico (polypropylene woven bags) in minimizing storage losses and maintaining seed germination was compared with hermetic storage technologies. After six and three months of storage, the percentage of insect damage and weight loss was highest in samples stored in polypropylene woven bags, reaching 61.4% and 23.4% after six months of storage. On the contrary, with hermetic technologies, storage losses were minimal, with maximums of 4.1% and 2.2% for insect damage and weight loss, respectively. Overall, the germination rate of samples stored in these airtight containers was greater than 90%. The results of this study demonstrate the potential of hermetic technologies in preserving the biodiversity of maize seed landraces and strengthening smallholders’ food security.

**Abstract:**

Smallholder farmers who grow maize landraces face important challenges to preserve their seed biodiversity from one season to another. This study was carried out in the central highlands of Mexico to compare the effectiveness of two seed storage practices—specifically, polypropylene woven bags (farmers’ conventional practice) vs. hermetic containers—for minimizing seed losses and maintaining germination. Four Mexican landraces were stored for three and six months. Data on moisture content and kernel damage were collected at the beginning and the end of the storage period. Pest-free samples collected were also analyzed for seed germination. Moisture content was below 13% overall and was not significantly affected by storage technology or storage time. Samples from the polypropylene woven bags suffered significant damage from *Sitophilus zeamais* and *Prostephanus truncatus*, with the percentages of insect damage and weight loss reaching 61.4% and 23.4%, respectively. Losses were minimal in seed stored in hermetic containers, with a maximum insect damage of 4.1% and weight loss of 2.2%. Overall, the germination rate of samples stored in these airtight containers was greater than 90%. This study provides additional evidence on the effectiveness of hermetic containers at maintaining Mexican landraces’ seed quantity and quality during storage in smallholder conditions in central Mexico.

## 1. Introduction

Mexico is a center of origin, domestication and diversity for maize, a major crop that contributes to the livelihoods of millions of people [1,2]. Maize landraces—heirloom varieties created through farmer selection over hundreds of years—are key to food sovereignty and cultural identity in Mexico and Latin America [3,4,5]. Smallholder farmers in Mexico grow landraces for food, as their grain quality is often preferred for popular dishes and they are sometimes better adapted to local environmental conditions and rainfed agriculture [5,6,7]. Farmers play an important role in preserving maize landrace diversity through their historical and cultural identification of maize as a primary crop, as well as their land use, selection of traits, and seed exchanges through social networks [8]. The in situ conservation of landrace diversity allows landraces to evolve in their original areas of distribution, influenced by farmers’ selection and environmental factors [5]. 

At the same time, maize landraces face serious threats, including global warming, low productivity, and high levels of postharvest loss under farmers’ conventional storage conditions [9,10]. Losses in the quantity and quality of grain and seed under storage have been estimated to be as high as 60% in lowland conditions under common on-farm storage technologies, which include the use of polypropylene woven bags with or without aluminum phosphide tablets [10]. Postharvest insect pests are the main cause of losses in dry maize, particularly the maize weevil *Sitophilus zeamais* (Motschulsky) (Coleoptera: Curculionidae), the large grain borer *Prostephanus truncatus* (Horn) (Coleoptera: Bostrichidae), and the Angoumois grain moth *Sitotroga cerealella* (Olivier) (Lepidoptera: Gelechiidae) [11]. Hermetically sealed grain storage technologies, including hermetic metal silos and hermetic bags, can minimize losses from storage pests [10,12], constraining or killing the insects through oxygen depletion and a subsequent increase in carbon dioxide [13]. However, hermetic containers are generally not available for many smallholders. Odjo et al. [10] demonstrated the effectiveness of other sealed storage alternatives in Mexico, including recycled containers (plastic bottles or plastic barrels) that may be a good fit for smallholder farmers for preserving grain and seed quantity and quality [10,14]. In a following study, Odjo et al. [14] also demonstrated that hermetic technologies including recycled containers can preserve seed quality in Mexico by maintaining a low-equilibrium relative humidity and seed dryness, as well as limiting oxygen availability during storage and minimizing grain quality loss.

The International Maize and Wheat Improvement Center (CIMMYT) has been promoting hermetic technologies for storage in smallholders’ farming systems in Mexico within the framework of the Integrated Agri-food Systems Initiative (IASI) [15,16]. This approach, implemented through the innovation hub model, goes beyond introducing innovation and has a strong focus on knowledge exchange and co-creation with farmers to achieve sustainable and systemic change [17]. During interactions with farmers, a decrease in the germination of maize landrace seeds stored in hermetic technologies was reported. Some local experts suggest the percussion of seeds before sowing, or allowing the seed to “breathe” outside the container for anywhere from three days to two weeks before sowing to increase germination rate [18]. 

A decrease in germination for seeds stored in hermetic containers has been reported for seeds with moisture content over 13% (wet basis). Adhikarinayake et al. [19] reported a decrease in seed viability in paddy rice stored under hermetic conditions at 14.1% moisture content, citing the combined effects of oxygen depletion and increased carbon dioxide and moisture content. Moreno-Martinez et al. [20] also reported lower germination in maize seed stored in hermetic containers at 15% moisture, citing the effect of low oxygen levels on seed embryos. Seed quality includes germination rate, viability, and longevity, and is generally affected by three main factors: seed moisture content, temperature, and the relative humidity of the storage environment. High oxygen levels are also associated with seed viability declines; mainly due to oxidative processes, particularly in seed stored at high moisture contents [21]. Storing low moisture content seed using hermetic technologies allows the preservation of seed quality by maintaining a low-equilibrium relative humidity and seed dryness, as well as limiting oxygen availability during storage [22,23,24,25]. However, there is a lack of evidence on the effect of different storage technologies, as well as the suggested seed treatments, on maize landrace seed germination. 

This study evaluates the effects of storage technologies (polypropylene woven bags vs. hermetic storage technologies) on maize landraces in the central highlands of Mexico, specifically, on major insect pests (*S. zeamais*, *P. truncatus*, and *S. cerealella*), kernel damage, and seed germination, as well as whether germination rate is affected by seed treatments (percussion and delaying sowing after opening hermetic technologies). 

## 2. Materials and Methods

### 2.1. Details of Experiments

On-farm storage experiments were established in 2019 at the experiment station of CIMMYT at El Batán Texcoco, the State of Mexico, Mexico (2282 m above sea level), for three and six months. Seeds of the four Mexican landraces used were sourced locally and grown on the station following identical agronomic practices (date of sowing, fertilization, pest managements, date of harvest). Harvested kernels were dried and stored as follows (Table 1): (1) polypropylene woven bags (PP) (farmers’ conventional storage practices) and (2) hermetically sealed containers, either the GrainPro Hermetic SuperGrainbag^®^ Premium RZ (GrainPro Inc, Washington, USA; bags obtained from the official Mexican representative), a hermetic bag with a zip system, or plastic bottles. GrainPro hermetic bags are made of high-strength polyethylene plastic with barrier layers, while plastic bottles are made of polyethylene terephthalate (PET). These materials have low oxygen permeability in comparison with polypropylene woven bags. Michiels et al. [26] reported the rate of dioxygen permeability measured at a relative humidity of 50% and a temperature of 23 °C for polypropylene materials between 50 and 100 cm^3^.mm.m^−2^.day^−1^.atm^−1^, while polyethylene plastics have values of dioxygen permeability between 0.5 and 5 cm^3^.mm.m^−2^.day^−1^.atm^−1^.

As reported by Odjo et al. [14], hermetic bags were used in combination with polypropylene woven bags as recommended by the provider for additional protection. Plastic bottles that were recycled water and soda containers were used for some of the varieties with low productivity, as farmers generally store 2 kg of seed. Both hermetic bags and plastic bottles were checked for any damage/perforation by inflating them. After filling with kernels, the hermetic bags were sealed with a zip system while the plastic bottles were sealed with their original cap wrapped with tape to minimize oxygen entry. The containers were randomly arranged on a wooden pallet platform with three replicates per storage technology for each storage time. 

The landraces used were of white, yellow, blue, and pink grain and with flotation indices (an indirect parameter of grain hardness determined by counting the number of floating grains after six strokes of mild stirring of 100 grains in a sodium nitrate solution with a specific density of 1.25 g mL^−1^) values varying from 34 to 80% (Table 1). Based on this parameter and the Mexican norm NMX-FF-034/1-SCFI-2002 [27], their endosperm hardness was intermediate (Magdalena and VC Rosado), hard (VC Amarillo) and soft (SJR Azul). Hermetic bags were used in comparison with polypropylene woven bags for Magdalena and VC Amarillo, while plastic bottles were used for VC Rosado and SJR Azul, since they have low productivity and farmers generally store around 2 kg of seed [28]. 

Before storage, moisture content was measured using a hand-held grain moisture tester (John Deere Moisture Check Plus Grain Moisture, IL, USA), calibrated using the manufacturer’s recommendations. Average moisture content varied between 10.0 and 12.6% on a wet basis (Table 1). After cleaning and before filling the storage containers, three representative samples of approximately 500 g were obtained to measure insect damage, referring to kernels with perforations or galleries caused by insect feeding. Insect damage before storage varied from 0.6 to 4.0% for the different varieties evaluated. The numbers of live insects that come from natural infestation in the field and were present in the samples were counted. The insect species considered are the most important in Mexico [11]: *S. zeamais*, *P. truncatus*, and *S. cerealella*. Generally, there was fewer than 1 live insect per 500 g of sample, prior to storage (Table 1). 

Three representative samples free of pests were also collected and stored at −18 °C for further seed analyses. After three and six months, the storage containers were opened, and three representative samples were collected from the top, middle, and bottom of each container using hand scoops. The seed moisture content and percentage of insect damage were measured using the method described above. The “weight loss” parameter, which refers to kernel loss during the storage period from insect feeding, was also estimated as per Boxall [29].

### 2.2. Germination Assays

Germination tests were performed on initial samples defrosted over one night at room temperature and samples without apparent damage collected after three and six months of storage, with or without percussion treatment. Samples without percussion were kept in open plastic jars at room temperature. 

Intended to create an abrasive effect on the seed pericarp and facilitate water imbibition, the percussion treatment was applied to approximately 100 g for 5 min just before the germination assay, using a device equipped with aluminum US standard testing sieve pans and adapted with a Power Electric motor, CPG1446RB1A (Mexico), at 1725 rpm, resulting in approximately 270 oscillations per minute. Germination assays were performed on the seeds obtained (no percussion, percussion) using the rolled paper towel seed germination test of CIMMYT [30] at 1, 3, 7, and 15 days after opening the storage containers. Fifty seeds without apparent damage were randomly selected and placed on the upper halves of moist filters or blotter paper towels (50 × 25 cm) and incubated in a seed incubator at 90% relative humidity, alternating between 12 h at 30 °C in light and 12 h at 20 °C in the dark, for 7 days, based on the 1985 recommendations of the International Seed Testing Association (ISTA). Germinated seeds were visually checked every day and “normal” and “abnormal seedlings” were counted by experienced lab technicians, meaning those that showed or did not show potential for continued development into satisfactory plants when grown in good soil under optimum conditions [30]. After 7 days, seeds that did not germinate were dissected longitudinally through the embryo and soaked in a 1% tetrazolium solution at 30 °C for 120 min as described by Warham et al. [30]. Through observation with an optical microscope, a seed is considered viable when at least 1/3 of the scutellum as well as the radicle are red-stained. The percentages of non-germinated viable and non-germinated non-viable seeds were determined with the counting of viable and non-viable seeds. All seed germination parameters (normal seedlings, abnormal seedlings, germination, non-germinated viable seeds, non-germinated non-viable seeds) were expressed as a percentage of all fifty seeds tested. 

### 2.3. Statistical Analyses

Statistical analyses of data collected were performed using R version 4.0.3. Data were first summarized as means and standard deviations. For the count data (numbers of live *S. zeamais*, *P. truncatus* and *S. cerealella*), a 95% Poisson confidence interval of the mean was estimated using the R package (“DescTools”). 

Beta regression was used to evaluate the effect of storage technologies, maize varieties, storage time, and their interactions on moisture content, percentage of insect damage, and weight loss, as well as to gauge the effect of storage technology, maize variety, storage time, seed treatment, number of days before the germination assay, and their interactions on the percentages of normal seedlings and abnormal seedlings [31]. All beta regressions were performed with the R package “mgcv” using the logit link as a link function. The parameters “storage technology”, “maize variety”, “storage time”, and “seed treatment” were considered as factors. The reference factors considered for each model were the “polypropylene woven bag” for storage technology, which is the common farmer practice; “Magdalena Texcoco” for maize variety, with white landraces being the area’s most popular maize variety; “three months of storage” for storage time and parameter, given that farmers store their harvests for at least three months [10]; and “no percussion” for the seed treatment parameter. Only two-way interactions have been considered within regression models to facilitate interpretation. 

Principal component analysis (PCA) was performed using the R package “FactoMineR” to evaluate relationships between postharvest parameters, including kernel moisture content and temperature at the end of the storage period, percentages of insect damage and kernels without damage, weight loss, numbers of live *S. zeamais*, live *P. truncatus* and live *S. cerealella*, and seed germination parameters (percentages of germination, normal seedlings, abnormal seedlings, non-germinated viable seeds, non-germinated non-viable seeds). Only final values for these parameters after 3 and 6 months of storage were used for the PCA.

## 3. Results

### 3.1. Effect of Storage Technologies on Live Counts of Insects

The main species of live insect counted was *P. truncatus*, followed by *S. zeamais* and *S. cerealella*. The highest numbers of insects were found in samples from the polypropylene woven bags, particularly after six months of storage. Magdalena and VC Rosado were the most highly infested maize varieties, particularly by *P. truncatus*, which was the most encountered living insect (Figure 1). The number of live insects of the three pest species in seed stored using hermetic containers varied between 0 and 2 insects per 500 g of seed, as shown by the 95% Poisson confidence intervals (Figure 1). In contrast, confidence intervals for seed stored using polypropylene woven bags were highly variable and depended on the maize variety and storage time. The upper estimates of the confidence intervals were as high as 9.5, 277.9, and 3.4 insects per 500 g of seed for *S. zeamais*, *P. truncatus* and *S. cerealella*, respectively (Figure 1).

### 3.2. Effect of Storage Technologies on Moisture Content, Insect Damage and Weight Loss

Kernel moisture content overall varied between 8.4 and 13.4% (Table 2). There were fluctuations between initial and final moisture content regardless of the storage technologies, particularly after six months of storage. Insect damage varied between 0.6 and 61.4%, while weight loss ranged from 0.2 to 20.1%, with heavy damage on kernels stored in polypropylene woven bags. Independently of the variety, insect damage was particularly severe after 6 months of storage and up to 14 times higher than after 3 months (and an average increase from 12.9 to 51.9% for polypropylene woven bags, with that increase in storage time). However, VC Amarillo, the variety with a hard endosperm, had the lowest insect damage and weight loss values after six months of storage using polypropylene woven bags while the most infested landraces were VC Rosado and SJR Azul. Kernel damage was particularly low with samples stored using hermetic technologies, whatever the variety or storage time, with weight losses from 0.2 to 1.2% and the same trend for insect damage. Hermetic storage minimized weight losses, whereas high weight losses occurred with polypropylene woven bags for all landraces, and particularly after 6 months of storage. 

### 3.3. Effect of Variety, Storage Time and Storage Technology on Normal, Abnormal Seedlings and Germination

Percentages of normal seedlings after 3 and 6 months of storage 1 day after opening the containers varied between 6.7 and 87.7%, while percentages of abnormal seedlings varied between 11.3 and 86.0% (Table 3), with these two parameters presenting opposite trends and a significant effect of variety. In some cases, the percentages of normal seedlings after storage were higher in comparison with the initial values. Overall, increasing storage time from 3 to 6 months had a negative impact on the percentage of normal seedlings (and the opposite for the percentage of abnormal seedlings). However, storing seed in hermetic containers positively impacted the percentage of normal seedlings after 6 months of storage, compared to storage using polypropylene woven bags (Table 3). Overall, applying percussion as a seed treatment did not have a significant effect (Table 3), but applying it after 6 months of storage seemed to have a negative impact on the percentage of normal seedlings. 

Delaying germination and aerating seed for 3, 7 or 15 days did not significantly affect the percentage of normal and abnormal seedlings (Appendix A). The percentages of non-germinated viable seeds and non-germinated non-viable seeds were very low, except in two cases when percussion was applied for grain stored using polypropylene woven bags (Appendix A). In general, post-storage seed germination rates were high (average > 90%) for all the storage technologies and maize landraces, regardless of seed treatment (Appendix A).

### 3.4. Relationship between Germination and Postharvest Loss Parameters

The PCA identified three components with eigenvalues >1. The first two principal components accounted for 66% of the total variability in results (Appendix A). The first component, which explained 48% of the total variability, was highly and positively correlated with insect damage, weight loss, and final numbers of live *S. zeamais*, *P. truncatus*, and *S. cerealella*. The first component was negatively correlated with percentage of germination. The second component, which represented 18% of the total variability, was positively correlated with percentage of normal seedlings and kernel temperature, and negatively correlated with percentage of abnormal seedlings (Appendix A). The number of live *S. zeamais*, weight loss, and the number of live *P. truncatus* are the parameters that contributed most to the first component, while percentages of normal and abnormal seedlings are the ones that contributed mainly to the second component (Appendix A). Overall, the loading plot of variables used for the PCA, using the first two components (Figure 2), showed that postharvest loss parameters contributed the most to component 1, while germination parameters contributed mainly to component 2. The loading plot also highlighted a close association between postharvest loss parameters and the percentage of non-germinated viable seeds. 

## 4. Discussion

Smallholder farmers in Mexico actively participate in the preservation of maize seed biodiversity through community seed banks [5,8,32]. Hermetic technologies can contribute to seed preservation by protecting the seed during the storage period. This study evaluated the effect of hermetic technologies on postharvest loss parameters and seed germination.

There was little variation in kernel moisture content across storage technologies or storage periods, though some significant increases and decreases were observed, particularly after six months of storage. Overall, kernels stored in either hermetic bags or sealed plastic containers had lower moisture content variation than those stored in polypropylene woven bags. These results are similar to those from other experiments in the State of Mexico published by García-Lara et al. [33], who reported an increase in grain moisture content stored using polypropylene sacks. The same results have also been reported for experiments elsewhere [34,35]. These changes are associated with environmental humidity and the failure of polypropylene woven bags to protect grain from external conditions, potentially increasing moisture content or drying stored grain, depending on the environment [36]. The fluctuations in moisture content noticed in hermetic bags, particularly after six months of storage were reported by other authors [24,37] and could be explained by ambient air leaking into the system. Hermetic bags are not perfectly “impermeable” and the sealing system (a zip system) of the storage containers could also have an impact on oxygen and moisture fluctuations within the technologies [38]. Fluctuations of moisture content could also be explained by the perforation of bags by insects, particularly *P. truncatus*, and/or biological activity inside the containers that may lead to condensation due to temperature variation outside the containers [13]. The perforation of hermetic bags by bruchids has been reported in experiments carried out elsewhere [39,40,41] even though the impact on insect damage was not significant. Overall, hermetic technologies with low-permeability barriers help to limit insect damage in comparison with non-hermetic technologies.

Samples stored in polypropylene woven bags were highly infested during storage, resulting in high levels of insect damage and weight loss, confirming previous findings for this region of Mexico [10,33]. Storage losses were due to the activities of the main pests, particularly *P. truncatus*, and favored by the availability of oxygen. However, VC Amarillo, the landrace with the hardest endosperm, seemed less infested by insects. The incidence of these pests in grain inside polypropylene woven bags depends on the landraces and their characteristics, including composition, kernel hardness and vitreousness, and pericarp thickness [42]. The endosperm characteristics are not the only mechanism involved in the resistance to storage insect pests. Previous studies have reported that maize resistance to postharvest insect pests is related to phenolic contents [43,44]. Maize kernel resistance to postharvest pests is based on complex interactions between anatomical, biochemical and genetic factors, and includes antibiosis and antixenosis (physical barriers and phytochemical repellents) effects [42]. Some Mexican landraces, particularly the ones of yellow, blue and pink colors, are known for their high polyphenol and anthocyanins contents [45]. These landraces with high resistance to the main postharvest pests have been identified and could be used to develop improved varieties [46]. However, since farmers’ selection of maize varieties also depends on their home consumption strategies [6], it is important to find other strategies to avoid losses during storage. Hermetic technologies, including low-cost alternatives such as recycled containers, e.g., plastic bottles and barrels, minimize insect infestations and weight loss regardless of maize variety in smallholders’ farming systems [10].

As highlighted by the PCA, there was a negative relationship between increased postharvest loss (high percentage of insect damage, weight loss, and high numbers of live insects) and seed germination. However, post-storage germination data of the pest-free samples stored in polypropylene woven bags were very high, and there was no significant effect of maize variety, storage technology, or storage time. Similar results were found by Odjo et al. [14], who reported high percentages of germination of seed stored using non-hermetic technologies in temperate conditions (>2000 masl). These results could be ascribed to environment, including low relative humidity and ambient temperature [10]. As per Harrington [21], storage conditions including low temperatures and low relative humidity strongly favor increased seed longevity. However, storage in polypropylene woven bags resulted in a loss of seed quantity, as demonstrated by the percentage of insect damage and weight loss recorded, so polypropylene woven bags are not a viable storage strategy for seed biodiversity preservation. Hermetic storage had no detrimental effects on the percentage of germination and applying percussion or delaying the germination assay did not significantly improve it. These results corroborate the findings of García-Lara et al. [33] and Odjo et al. [14] in the central highlands of Mexico and experiments elsewhere [34,47,48,49]. Hermetic technologies preserve the percentage of seed germination by limiting oxygen availability and maintaining low-equilibrium relative humidity for stored seed, which in turn slows oxidative reactions associated with decreased seed viability [50]. These results are valid over an extended storage time (up to 7 months), as demonstrated by Kuyu et al. [25]. Regarding special treatments, percussion treatments in this study tended to reduce the percentage of normal seedlings. No special treatments were required for a satisfactory germination rate in the case of the maize varieties studied.

A decreased germination rate for seed stored using hermetic technologies has been reported and may be associated with the conditions of germination assays, particularly seed moisture content [19,20,51]. Singano et al. [51] reported a low percentage of seed germination (less than 15%) for samples stored using hermetic technologies and associated with high temperatures and moisture content and the airtight conditions. Water availability in hermetic storage can provoke fermentation in the seed, decreasing viability [52]. Seed moisture content at storage in the current study was below 13%, explaining the high germination recorded, and the data presented here confirm that storing dry seed in hermetically sealed containers will not adversely affect seed germination.

The percentages of normal seedlings reported in the current study were low. Landraces are generally selected by farmers in Mexico based on ear corn and kernel characteristics for culinary qualities and cultural purposes [1,6]. Participatory breeding programs considering farmers’ preferences could help in improving landraces’ seed quality and overall productivity, and enhance their in situ conservation [5]. The percentage of normal seedlings for initial samples frozen at −18 °C were sometimes lower than normal seedling values obtained after storage, suggesting freezing injury. This phenomenon has already been described during the storage of high-moisture-content seeds (>15%) [50]. The results presented in the current study suggest that freezing injury could still happen at moisture contents between 10.0 and 12.6% (the moisture contents of the initial materials). Data on the effect of freezing on Mexican seed landraces quality is, however, scarce, and this warrants further investigation. A decrease in the percentage of normal seedlings of seed stored in hermetic containers was sometimes observed and, overall, storage time also had a negative effect on the percentage of normal seedlings. These results are unexpected and contrast with hybrid seeds’ germination results from the same area presented by Odjo et al. [14]. This could be associated with seed moisture content, and the results presented in the current study suggest that an additional decrease in moisture content for the landraces studied is required. Seed preservation standards generally recommend drying seed to a critical moisture content below which desiccation injury could occur [53]. According to the results of Bakhtavar et al. [54], only seed with 8% moisture content stored in hermetic containers maintained a satisfactory germination level. Adsorption isotherm studies of these landraces could help to establish the best conditions for their storage. Overall, seed drying, a key preparation prior to storage in hermetic conditions, is important but could be challenging for smallholders, particularly in the highlands or during rainy seasons [55]. Sun drying, particularly at hot times of day, can also reduce seed germination. Low-cost seed dryers or shade drying can be used without detrimental effects on germination [56]. Solutions such as zeolite drying are also safer alternatives that can maintain maize seeds’ high viability during storage. These drying solutions combined with the dry chain strategy have been promoted for Guatemala’s community seed reserves to preserve seed viability [57].

## 5. Conclusions

In the case of maize stored for use as seed by smallholder farmers, hermetic technologies can limit damage by postharvest insect pests and minimize storage loss. Quantitative damage was much higher with polypropylene woven bags with high infestation levels of *S. zeamais* and *P. truncatus*. Using hermetic technologies, including recycled storage containers such as plastic bottles, did not significantly affect germination rate, and there was no need for additional seed treatments to increase the percentage of germination. These technologies could be promoted for smallholder farmers in Mexico with the appropriate technical support, including properly drying seeds before storage. Facilitating the physical and economical access of smallholders to these technologies and practices so that they can dry and store seed in safe conditions is critical to preserve their seed biodiversity and strength their food security.

## Figures and Tables

**Figure 1 insects-13-00878-f001:**
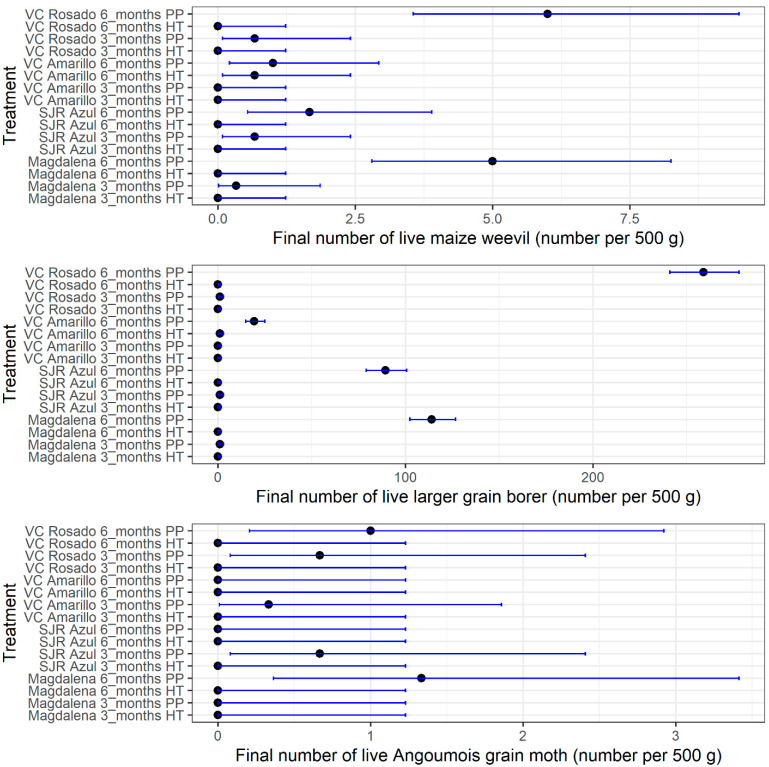
Poisson confidence intervals (95%) of the mean of the final number of live maize weevils (*S. zeamais*), larger grain borers (*P. truncatus*) and Angoumois grain moths (*S. cerealella*) after three and six months of storage. Legend. PP: polypropylene woven bag; HT: hermetic technologies.

**Figure 2 insects-13-00878-f002:**
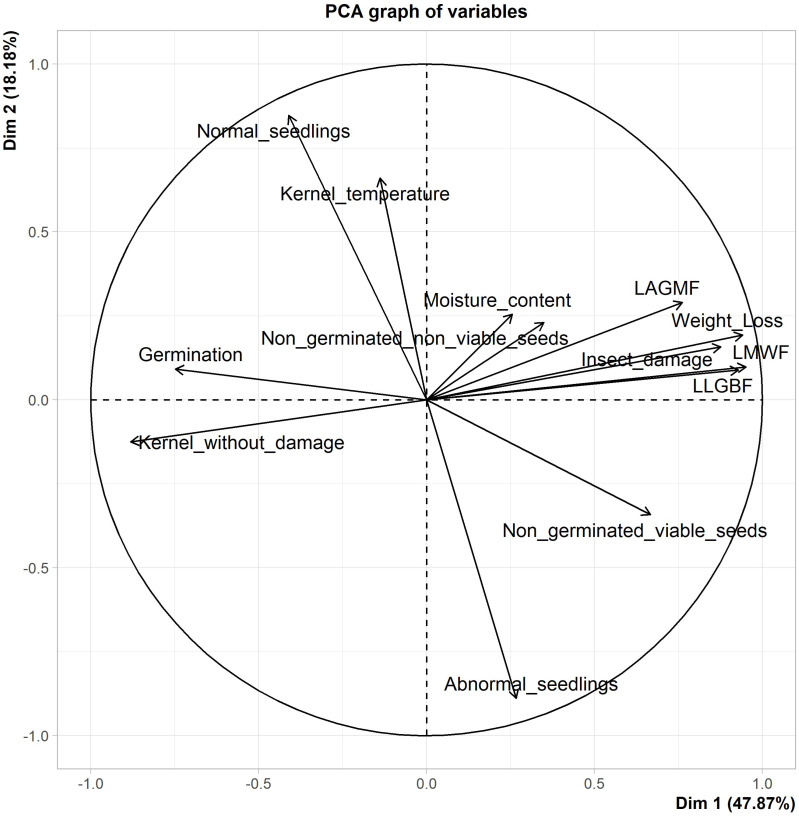
Principal component analysis loading plot with postharvest loss and germination parameters after storage. Legend: Insect_damage: percentage of insect damage; Weight_ loss: weight loss; LMWF: final number of live maize weevil; LLGBF: final number of live larger grain borer; LAGMF: final number of live Angoumois grain moth; Kernel_without_damage: percentage of kernel without damage; Moisture content: grain moisture content; Kernel_temperature: kernel temperature; Germination: percentage of germination; Normal_seedlings: percentage of normal seedlings; abnormal_seedlings: percentage of abnormal seedlings; Non_germinated_viable_seeds: percentage of non-germinated viable seeds; Non_germinated_non_viable_seeds: percentage of non-germinated non-viable seeds.

**Table 1 insects-13-00878-t001:** Average (±standard deviation) damage parameters at the beginning of the experiment.

Parameters	Landraces			
	Magdalena	VC Amarillo	VC Rosado	SJR Azul
Color	White	Yellow	Pink	Blue
Flotation index (*n* = 3)	48.2 ± 3.55	34.2 ± 2.02	49.5 ± 3.50	80.5 ± 5.22
Endosperm hardness classification ^1^	Intermediate	Hard	Intermediate	Soft
Moisture content (%, wet basis) (*n* = 3)	10.0 ± 0.53	10.5 ± 2.86	12.6 ± 0.06	11.8 ± 0.49
Temperature (°C) (*n* = 3)	22.2 ± 0.06	21.5 ± 0.10	21.6 ± 0.15	21.4 ± 0.08
Insect damage (%) (*n* = 3)	0.6 ± 0.37	1.7 ± 0.19	2.7 ± 0.33	4.0 ± 1.01
Live *S. zeamais* (number per 500 g) (*n* = 3)	0	1.0 ±1.73	0.3 ± 0.58	0.3 ± 0.58
Live *P. truncatus* (number per 500 g) (*n* = 3)	0	0	0	0.3 ± 0.58
Live *S. cerealella* (number per 500 g) (*n* = 3)	0	0	0	0
Storage technologies evaluated	PP, HBZ	PP, HBZ	PP, PBO	PP, PBO

Legend: PP: polypropylene woven bag; PBO: plastic bottle; HBZ: GrainPro hermetic SuperGrainbag^®^ Premium RZ with zip. ^1^ Endosperm harness classification according to the Mexican norm (NMX-FF-034/1-SCFI-2002) using the flotation index value [27].

**Table 2 insects-13-00878-t002:** Average moisture content and damage (±standard deviation) after 3 and 6 months of storage.

Maize Variety	Storage Time (Months)	Storage Technology	Moisture Content (%)		Insect Damage (%)		Weight Loss (%)	
Magdalena	0	INI	10.0 ± 0.53		0.6 ± 0.37		-	
	3	PP	11.4 ± 0.03		3.5 ± 0.65		0.5 ± 0.22	
	3	HT	9.5 ± 0.46		1.3 ± 0.33		0.2 ± 0.06	
	6	PP	13.4 ± 0.95		50.5 ± 12.19		20.1 ± 8.51	
	6	HT	11.3 ± 0.40		1.2 ± 0.76		0.2 ± 0.17	
C Amarillo	0	INI	10.5 ± 2.86		1.7 ± 0.19		-	
	3	PP	11.7 ± 0.29		2.4 ± 0.15		0.5 ± 0.04	
	3	HT	8.4 ± 0.05		1.3 ± 0.45		0.2 ± 0.06	
	6	PP	9.5 ± 0.36		36.2 ± 2.83		5.7 ± 4.30	
	6	HT	9.5 ± 0.15		2.2 ± 2.16		0.6 ± 0.61	
VC Rosado	0	INI	12.6 ± 0.06		2.7 ± 0.33		-	
	3	PP	10.9 ± 0.28		38.1 ± 5.50		12.8 ± 1.85	
	3	HT	11.5 ± 0.38		4.1 ± 1.22		0.5 ± 0.35	
	6	PP	10.4 ± 0.42		61.4 ± 8.78		23.4 ± 1.90	
	6	HT	8.5 ± 0.00		2.4 ± 0.7		0.6 ± 0.36	
SJR Azul	0	INI	11.8 ± 0.49		4.0 ± 1.01		-	
	3	PP	11.0 ± 0.39		7.4 ± 0.98		2.2 ± 0.80	
	3	HT	11.7 ± 0.15		3.8 ± 1.43		2.2 ± 1.12	
	6	PP	9.2 ± 0.10		59.5 ± 1.26		15.0 ± 6.20	
	**6**	HT	12.1 ± 0.20		3.1 ± 1.14		1.17 ± 0.28	
Statistical analyses								
Independent variables			Moisture content (%)		Insect damage (%)		Weight loss (%)	
			Estimate	*p*	Estimate	*p*	Estimate	*p*
Intercept			−2.002	<0.001 ***	−2.900	<0.001 ***	−4.357	<0.001 ***
HT			−0.305	<0.001 ***	−1.381	0.001 **	−1.591	0.001 **
SJR Azul			−0.139	0.019 *	0.475	0.216	0.795	0.089
VC Amarillo			−0.090	0.129	−0.389	0.380	−0.385	0.484
VC Rosado			−0.028	0.634	2.356	<0.001 ***	2.402	<0.001 ***
Six months			0.092	0.087	2.874	<0.001 ***	2.885	<0.001 ***
HT × SJR Azul			0.475	<0.001 ***	0.768	0.088	1.405	0.006 **
HT × VC Amarillo			0.097	0.173	0.667	0.192	1.238	0.038 *
HT × VC Rosado			0.217	0.002 **	−0.824	0.081	−0.905	0.101
HT × Six months			0.036	0.470	−2.278	<0.001 ***	−1.701	<0.001 ***
SJR Azul × Six months			−0.183	0.008 **	−0.070	0.866	−1.111	0.023 *
VC Amarillo × Six months			−0.168	0.018 *	−0.160	0.734	−0.985	0.090
VC Rosado × Six months			−0.293	<0.001 ***	−1.800	<0.001 ***	−2.065	<0.001 ***
Model’s parameters								
Adjusted R^2^			0.67		0.941		0.863	
Deviance Explained (%)			75.2		96.4		93.8	
N			48		48		48	

Legend: INI: initial samples; PP: polypropylene woven bag; HT: hermetic technologies; R^2^: coefficient of determination; N: number of observations; *: *p* < 0.05; **: *p* < 0.01; ***: *p* < 0.001.

**Table 3 insects-13-00878-t003:** Average percentages (±standard deviation) of normal seedlings and abnormal seedlings, after 3 and 6 months of storage with germination assays performed 1 day after opening containers.

Maize Variety	Storage Time (Months)	Storage Technology	Normal Seedlings (%)		Abnormal Seedlings (%)	
			No Percussion	Percussion	No Percussion	Percussion
Magdalena	0	INI	47.3 ± 6.11	47.3 ± 6.11	44.7 ± 4.16	44.7 ± 4.16
	3	PP	58.7 ± 3.05	64.7 ± 1.15	37.3 ± 4.16	32.0 ± 3.46
	3	HT	58.7 ± 3.05	58.7 ± 2.31	28.0 ± 6.00	32.0 ± 0.00
	6	PP	47.3 ± 12.86	16.0 ± 7.21	43.3 ± 9.02	76.7 ± 10.07
	6	HT	22.0 ± 6.00	56.7 ± 6.11	73.3 ± 7.57	37.3 ± 8.08
VC Amarillo	0	INI	72.7 ± 7.02	72.7 ± 7.02	26.0 ± 7.21	26.0 ± 7.21
	3	PP	83.3 ± 1.15	75.3 ± 11.72	16.0 ± 0.00	24.7 ± 11.72
	3	HT	80.0 ± 9.17	80.0 ± 9.17	16.7 ± 6.11	18.0 ± 8.00
	6	PP	39.3 ± 8.08	10.0 ± 2.00	57.3 ± 7.57	86.0 ± 2.00
	6	HT	76.7 ± 5.03	66.0 ± 2.00	22.0 ± 4.00	32.7 ± 1.15
VC Rosado	0	INI	67.3 ± 8.08	67.3 ± 8.33	28.7 ± 8.33	28.7 ± 8.33
	3	PP	72.0 ± 0.00	87.7 ± 5.51	24.3 ± 3.79	11.3 ± 5.03
	3	HT	78.0 ± 5.57	77.3 ± 5.51	19.3 ± 5.51	16.3 ± 6.03
	6	PP	40.0 ± 8.72	56.7 ± 8.33	50.0 ± 5.29	36.0 ± 8.00
	6	HT	58.7 ± 4.16	16.0 ± 4.00	38.7 ± 2.31	76.7 ± 3.06
SJR Azul	0	INI	60.0 ± 3.46	60.0 ± 3.46	36.0 ± 2.00	36.0 ± 2.00
	3	PP	73.7 ± 11.68	78.3 ± 8.08	22.3 ± 10.69	17.7 ± 6.43
	3	HT	78.3 ± 7.51	68.0 ± 7.81	19.0 ± 7.81	27.0 ± 8.72
	6	PP	56.7 ± 8.08	6.7 ± 4.16	33.3 ± 4.16	84.0 ± 4.00
	6	HT	62.0 ± 2.00	93.3 ± 1.15	33.3 ± 3.06	51.3 ± 4.16
Statistical analyses						
Independent variables			Normal seedlings (%)		Abnormal seedlings (%)	
			Estimate	*p*	Estimate	*p*
Intercept			0.351	0.166	−0.594	0.014 *
HT			−0.402	0.165	0.166	0.544
SJR Azul			0.988	0.004 **	−0.943	0.004 **
VC Amarillo			0.768	0.024 *	−0.586	0.070
VC Rosado			1.293	<0.001 ***	−1.169	<0.001 ***
Six months			−1.036	<0.001 ***	0.937	<0.001 ***
Percussion			0.367	0.208	−0.261	0.345
HT × SJR Azul			0.323	0.342	−0.079	0.804
HT × VC Amarillo			0.992	0.004 **	−0.930	0.004 **
HT × VC Rosado			−0.514	0.131	0.630	0.049 *
HT × Six months			0.793	0.001 **	−0.465	0.046 *
HT × Percussion			0.217	0.376	−0.257	0.267
SJR Azul × Six months			−0.457	0.180	0.276	0.390
VC Amarillo × Six months			−0.429	0.215	0.416	0.204
VC Rosado × Six months			−0.609	0.074	0.487	0.132
SJR Azul × Percussion			−0.979	0.004 **	0.841	0.009 **
VC Amarillo × Percussion			−0.637	0.065	0.600	0.065
VC Rosado × Percussion			−0.253	0.456	0.120	0.709
Six months × Percussion			−0.778	0.002 **	0.673	0.004 **
Model’s parameters						
Adjusted R^2^			0.690		0.678	
Deviance Explained (%)			76.8%		74.7%	
N			96		96	

Legend: INI: initial samples frozen; PP: polypropylene woven bag; HT: hermetic technologies; R^2^: coefficient of determination; N: number of observations; *: *p* < 0.05; **: *p* < 0.01; ***: *p* < 0.001.

## Data Availability

The data presented in this study are openly available in DataVerse: CIMMYT Research Data & Software Repository Network, V1. Verhulst, Nele; Odjo, Sylvanus; Palacios, Natalia, 2022, “Dataset of grain damage and seed quality of maize stored for three and six months in the Central Highlands of Mexico in 2019”, https://hdl.handle.net/11529/10548766 (accessed on 20 July 2022).

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
