# Peer review of "Effect of Storage Technologies on Postharvest Insect Pest Control and Seed Germination in Mexican Maize Landraces"

_insects, 2022, doi:10.3390/insects13100878_

Round 1

Reviewer 1 Report

Few comments regarding the paper

In row 40 used is expression “Climate change”. If it should be thought that high temperatures are a reason for poor storage conditions and damages during storage better expression will be “Global warming”

Is in row 66 the second “of” too much?

In row 107 one bracket “)” is too much

In the row 172 “Olivier” is not needed

I believe that in Table 2 missing is storage time for some storage technology(ies) (like in Table 3)

If it is available it will be good to have information about characteristics of PP bags about permeability for air, moisture etc… because this is the reason for poor storing conditions and damages.

It would be important to explain how plastic bottles are sealed to obtain hermetic storage and if this bottles are available and where.

The paper is worth publishing due to the information that PP bags that are used by farmers could be replaced by sealed plastic bottles that give better results, especially lower insect damage.

Recommendation for the future work: Study possible use of PolyamidePolyethylene (PA/PE) bags for better preservation of stored seeds and possible use of carbon dioxide

Reviewer 2 Report

The work is good mainly for a statistical analysis point of view in using different statistical tools for determining parameters. Several similar jobs have been done, and it is good if results are compared and contrasted with others' work by focusing more on the technical aspects. Several justifications need to be included in the discussion part in line with the results reported in the result part. 

-Land races are not equally subjected to similar packaging technologies; please refer to table 1, and no justification is given. 

- Results in table 3 seem to contradict when compared with initial values. How are values after six months higher than zero time?

- Please try to address all the components in the result part in the discussion part too. 

- Please refer comments in the attached document. 

Reviewer 3 Report

General comments:

1-      This study has tried to enlighten the effect of various storage technologies on maize germination. Among the tested technologies was hermetic storage. In hermetic storage, the basis is to have a well-sealed enclosure. As it has been shown in the literature, leaking storage cannot be considered hermetic storage. In this study, three types of hermetic storage structures were tested: GrainPro Hermetic SuperGrainbag® Premium RZ, a hermetic bag with a Zip system (HBZ), or plastic bottles (PBO). We have no information on their degree of gas-tightness, whether they were tested for their gas-tightness and what was their oxygen permeability rate (OTR). This information is crucial for the success or failure of hermetic storage technology. The authors should indicate whether each structure was tested for its gas-tightness before the start of the storage period. In addition, they should indicate the permeability level in OTR units.

2-     Changes in moisture content under hermetic conditions: it has been well established that for dry grains there is no change in moisture content over extended periods of storage under hermetic conditions. Table 2 gives fluctuating results over 3 and 6 months of storage. For example, Magdalena variety under HT at 3 months is 9.5% and after 6 months is 11.3%. This is not a negligible change that needs explanation. Is this an experimental error or a true increase in moisture content? It is crucial that those results of fluctuating moisture contents over time be explained adequately.

3-     Germination values given in Table 3 are very interesting. First of all initial germination values are very low. Is this common and acceptable for the Mexican varieties? In other countries germination values below 95% are not acceptable. Authors should explain the source of such low germination values in all tests. Is this because those were already damaged during drying or due to other reasons? Those low germination values need discussion.

4-     Under HT germination values for Magdalena variety dropped from 58.7% to 22.0%. This is a very significant decrease. The reason for such a decrease in germination under HT needs an explanation in the discussion section of the manuscript. The effect of hermetic storage on seed germination has been very well documented [Navarro, S. (2006) Modified Atmospheres for the Control of Stored-Product Insects and Mites. In: Insect Management for Food Storage and Processing, Second Edition. Heaps, J. W. Ed., AACC International, St. Paul, MN, pp. 105-146.]. A decrease in germination occurs only if the moisture content is above its critical storability limits. This aspect of reduced germination under HT should be seriously addressed in the discussion section of the manuscript.

Specific comments:

1-     Lines 54-57: the sentence "… but the impact of sealed storage technologies on seed quality is not well documented and variable effects on viability have been reported from studies using seed stored under low oxygen concentrations [15]. " is not correct. The quotation of the reference Groot et al (2015) is a laboratory work on seed longevity. It is well established that hermetic storage is an excellent solution for preserving seed quality. Authors must make additional efforts to find out more relevant references.

2-     Lines 68-74, the authors are quoting works carried out using paddy at moisture contents that are well above the critical limits of storability as seed. Study of Adhikarinayake et al (2006) was carried out on paddy of 14.1% moisture content (wet basis) which is equivalent to 73% equilibrium relative humidity (ERH). While for seed storage the ERH should be well below 60%. The other reference they quoted was of Moreno-Martinez et al (2000) that studied maize of 15% moisture content (wet basis) which is equivalent to 78% equilibrium relative humidity (ERH). Those are ERH values that promote intensive respiration due to mold activity. Authors totally ignore that the basic rule of seed storage is low ERH. High ERH values as in those studies indicated by the authors are unacceptable for seed storage whether under hermetic or non-hermetic storage. Therefore, the quotation of those references is misleading and needs to be corrected by adding other references that give values at low ERH values [Navarro, S., Donahaye, E. and Fishman Svetlana (1994) The future of hermetic storage of dry grains in tropical and subtropical climates. Proceedings of the 6th International Working Conference on Stored-Product Protection, (Edited by Highley, E., Wright, E.J., Banks, H.J. and Champ, B.R.), Canberra, Australia, 17-23 April 1994, CAB International, Wallingford, Oxon, UK, pp 130-138].  

3-     Lines 75-78: Authors quote informal reports of maize farmers in Mexico. Those are based on rumors rather than scientific detailed studies. You may quote maize farmers but add that the effect of percussion and allowing the seed to breathe has not been studied scientifically.

4-     Lines 80-81: seed quality cannot be improved during storage. The only improvements are related to improving storage conditions by reducing the temperature or the moisture content. But if the seed has initially low germination power there are no studies that indicate improvement during storage. The attempt is to improve the storability conditions. This sentence needs revision.

5-     Line 101: average moisture content should be indicated that it is on wet basis.

6-     Table 2: There are missing values for storage times.

7-     Table 2 and Table 3: You may guess the data for each maize variety, but better buy allows a space among each variety or draws a separation line.

8-     Line: 262-264: The sentence "… There was little variation in kernel moisture content across storage technologies or storage periods, though some significant increases were occasionally observed in the case of non-hermetic containers…" needs improvement in the background of the comments made above. The sentence needs to refer to the reason for the fluctuating moisture contents results over time under hermetic conditions.    

Round 2

Author Response

We made some additional changes to the manuscript to clarify context and improve the discussion, in response to specific comments of reviewer 3. Reviewer 2 did not give any specific comments in this second round (just a general note that references could be improved), so we do not have a more specific response.

Reviewer 3 Report

This manuscript is a revised version of the study on "Effect of storage technologies on postharvest pest control and seed germination in Mexican maize landraces". The objective of this study was to evaluate the effects of storage technologies on maize landraces in the Central Highlands of Mexico, specifically on insect pests, kernel damage, and seed germination, as well as whether seed quality can be improved through seed treatments after storage. This study was carefully planned and performed and advanced scientific methods were used to analyze the results.

General comments:

1-      The new revised version of the study is accompanied by serious scientific errors.

2-     Authors insist on trying to rely on statistical analysis where the experimental errors indicate serious deviation from the logic. As it was explained in my previous review, changes in moisture content under hermetic conditions have been well established that for dry grains there is no change in moisture content over extended periods of storage under hermetic conditions. Table 2 gives fluctuating results over 3 and 6 months of storage. For example, Magdalena variety under HT at 3 months is 9.5% and after 6 months is 11.3%. This is an increase of 18.9% that cannot be tolerated. To reach such a level of moisture there should be a significant number of perforations in the hermetic liner while the package is exposed to high humidity. On the other hand for VC Rosado the initial moisture content of 12.6% dropped to 8.5% after 6 months under hermetic conditions. This is a drying effect of 32.5%. We do not know not if the storage was under wet or dry ambient relative humidity conditions. For the Magdalena variety, there was an increase and for VC Rosado there was a drying effect. Is this an experimental error or a true increase in moisture content? Those changes in moisture content were not explained adequately.

3-     The variations in germination values given in Table 3 were not explained adequately. . Although the authors explain the source of low initial germination values, they failed to explain the increase in germination or significant decrease in germination under hermetic storage. For Magdalena variety the initial is 47.3%, after 3 months is 58.7% (increase) and after 6 months is 22.0% (decrease) for maize of initial 10.0% moisture content. What could bring the germination to drop to such a low level?

The reason for such a decrease in germination under HT needs an explanation in the discussion section of the manuscript. The effect of hermetic storage on seed germination has been very well documented [Navarro, S. (2006) Modified Atmospheres for the Control of Stored-Product Insects and Mites. In: Insect Management for Food Storage and Processing, Second Edition. Heaps, J. W. Ed., AACC International, St. Paul, MN, pp. 105-146.]. A decrease in germination occurs only if the moisture content is above its critical storability limits. This aspect of reduced germination under HT was not sufficiently addressed in the discussion section of the manuscript.

Specific comments:

Line  14: "   Sitophilus zeamais"  and "Prostephanus truncatus"  Italics.

Lines 59-60: "… Postharvest insect pests are the main cause of losses, particularly…" should be "… Postharvest insect pests are the main cause of losses in dry maize, particularly…"

Lines 76-83: "This approach… germination rate [18]" those 7 lines should be deleted. The mentioned approach has nothing to do with hermetic storage technology. The cited references [17] and [18] are not relevant to this discussion.

Lines 97-99: "However,…..seed germination."  Should be deleted. Mexican landraces are no different than other maize varieties concerning germination. This sentence is not correct.

Lines 120-121: "…polypropylene bags…" should be "…polypropylene woven bags…"

Line 135: "…polypropylene bags…" should be "…polypropylene woven bags…"

Line 240: "…polypropylene bags…" should be "…polypropylene woven bags…"

Line 271: "… deviation) of germination percentages and normal seedlings,…" should be "… deviation) germination percentages of normal and abnormal seedlings,…"

Table 3: "No treatment" should be "No percussion"

Line 318: "…moisture content sometimes noticed…" should be "…moisture content noticed…"

Line 319-326: "…24,38] could be explained by perforation of bags…to limit insect damage." In this discussion, only exposure of the maize to ambient air due to insect perforations is considered. However, as in the general comments number 2 above, experimental errors also might have taken place in this study, which should be included in this discussion.

Line 324: "… from the air…"  should be "… from the ambient air…".

Line 327: "…polypropylene bags…" should be "…polypropylene woven bags…"

Line 330: "… P. truncatus .." Italics.

Line 332: "…polypropylene bags…" should be "…polypropylene woven bags…"

Lines 394-395: "...Bakhtavar et al. [53] suggest to dry seed at 8% moisture content for the preservation of seed quality in hermetic storage…" this sentence is not correct. Bakhtavar et al. did not suggest. They just tested 8% moisture content and found it appropriate. There is a big difference between suggested and tested. Please correct this sentence or delete it.

Line 406: "…polypropylene bags…" should be "…polypropylene woven bags…"

Author Response

see attached response
